# UnoLoRA: Single Low-Rank Adaptation for Efficient Multitask Fine-tuning

## Abstract

Recent advances in Parameter-Efficient Fine-Tuning (PEFT) have shown Low-Rank Adaptation (LoRA) to be an effective implicit regularizer for large language models. Building on these findings, we propose UnoLoRA, a novel approach that leverages a single shared LoRA module for efficient multi-task learning. While existing methods typically use separate LoRA adaptations for each task, our approach demonstrates that a single shared adapter can effectively capture both task-specific and task-agnostic knowledge. We further introduce UnoLoRA$^\star$, an enhanced variant that employs a shared hypernetwork to generate task-specific embeddings, improving convergence and task adaptation. Our method significantly reduces trainable parameters to just 0.05% per task while maintaining competitive performance on the GLUE benchmark. Our analysis reveals that the A and B matrices in our shared LoRA adapter naturally develop complementary roles: A matrices capture generalizable features across tasks, while B matrices specialize in task-specific representations. Our results show that sharing a single LoRA adapter can achieve efficient multi-task learning while significantly reducing memory requirements, making it particularly valuable for resource-constrained applications.

## 1 Introduction

The recent progress of Large Language Models (LLMs) has advanced the field of Natural Language Processing significantly, but their increasing sizes make deployment and adaptation for specific or multiple tasks complicated, making parameter-efficient methods essential. Multi-task learning is advantageous in several ways, such as helping models develop robust and transferable representations, lowering memory usage, and making it easier to adapt to multiple new tasks. However, it comes with a set of challenges, including negative transfer, where learning one task can hurt the performance of the model on other tasks, and the need for more model parameters, which can reduce efficiency.

Parameter-Efficient Fine-Tuning (PEFT) methods, particularly Low-Rank Adaptation (LoRA) (Hu et al., 2021), which our work builds upon, have gained attention due to their ability to adapt models to new tasks with minimal overhead. Recent studies have shown that LoRA behaves as an implicit regularizer (Biderman et al., 2024), helping mitigate catastrophic forgetting and maintaining diverse generations, suggesting its suitability for multi-task learning.

Building on these insights, we introduce UnoLoRA, a novel approach that uses a single LoRA module for efficient multi-task learning in LLMs. Unlike previous methods that use separate LoRA adapters for each task, UnoLoRA employs a shared LoRA module across all the tasks, capitalizing on LoRA's suspected implicit regularization properties to facilitate knowledge sharing effectively between tasks.

This addresses some of the challenges in multi-task learning - it ensures parameter efficiency by using a single low-rank adaptation, minimizing the additional parameters required for multiple tasks. The shared nature of the LoRA module allows for task-agnostic adaptations that mitigate negative transfer between tasks.

Furthermore, we introduce UnoLoRA$^\star$, which uses a shared hypernetwork to generate task-specific embeddings on top of UnoLoRA, which allows the model to distinguish between different tasks and learn task-specific adaptations.

We evaluate UnoLoRA and UnoLoRA$^\star$ on the GLUE benchmark (Wang, 2018), demonstrating their effectiveness in a multi-task setting. Our experiments show that UnoLoRA$^\star$ achieves competitive

performance with existing multi-task approaches on GLUE while offering improved parameter efficiency and having a significantly higher per-step convergence than UnoLoRA.

The main contributions of our work are:

1. A novel single-LoRA-based architecture designed for multi-task learning in LLMs.
2. Comprehensive empirical evaluation of UnoLoRA and UnoLoRA$^\star$ on GLUE.
3. Analysis and visualization of the behavior and properties of LoRA matrices in single-task versus multi-task settings.

## 2 RELATED WORK

### 2.1 MULTI-TASK LEARNING IN LARGE LANGUAGE MODELS

Multi-task learning (MTL) in Large Language Models (LLMs) has gained attention due to the potential for improving model generalization and efficiency. Older approaches often involve full fine-tuning the model on multiple tasks simultaneously (Liu et al., 2019; Aghajanyan et al., 2021). This can lead to challenges such as negative transfer and increased computational requirements (Wang et al., 2019).

Recent work has explored more parameter-efficient approaches. Adapter based methods (Houlsby et al., 2019; Pfeiffer et al., 2020) introduce small task specific modules while keeping the base model frozen. Prompt-tuning techniques (Lester et al., 2021; Li & Liang, 2021) modify the input representation to adapt models to new tasks.

A significant advancement in this area is the HyperFormer approach, introduced by Mahabadi et al. (2021). This method employs shared hypernetworks for parameter-efficient multi-task fine-tuning of Transformers. HyperFormer learns to generate task specific adapter parameters, enabling efficient sharing of knowledge across tasks while maintaining task specific adaptations. This approach significantly reduces the number of per task trainable parameters compared to traditional adapter methods, while achieving superior performance on GLUE.

### 2.2 IMPLICIT REGULARIZATION IN NEURAL NETWORKS

Regularization in machine learning is essential in preventing overfitting and improving model generalization. Implicit regularization, which refers to the natural biases of optimization methods or architectural constraints towards simpler, more generalizable solutions (Neyshabur et al., 2017), has gained attention in deep learning. Implicit regularization happens without explicit regularization terms, as seen in several training dynamics and optimization algorithms (Gunasekar et al., 2017).

Recent work shows that Low-Rank Adaptation (LoRA) possesses strong implicit regularization properties. Biderman et al. (2024) found that LoRA's low-rank structure leads to learning less and forgetting less compared to full fine-tuning, suggesting it constrains models in ways that mitigate catastrophic forgetting and may promote positive transfer between tasks.

LoRA's regularization aligns with broader trends in deep learning, where limiting the parameter space tends to improve generalization. Techniques like pruning (Han et al., 2015) and quantization (Jacob et al., 2018) achieve model complexity reduction while maintaining performance through implicit regularization.

LoRA's parameter efficiency proves to be effective in the multi-task learning scenario. By sharing a single LoRA module across multiple tasks, its regularization properties enable effective fine-tuning, balancing task-specific adaptations with general language understanding. This shared adaptation of LoRA both prevents overfitting and promotes generalization. This allows UnoLoRA$^\star$ to deliver competitive performance with minimal additional parameters.

## 3 METHODOLOGY

**Problem Formulation.** Our focus is on a multi-task learning problem where we seek to develop a single model capable of performing well across diverse tasks. Consider a pre-trained language model

$M_\theta$ with parameter set $\theta$, and a collection of target tasks $\mathcal{T} = \{T_1, T_2, \ldots, T_K\}$, where $K$ represents the total number of tasks. Our goal is to determine an optimized set of parameters $\theta^*$ that maximizes performance across all the tasks.

For each task $T_j \in \mathcal{T}$, we have a corresponding dataset $D_j = \{(X_j^i, Y_j^i)\}_{i=1}^{M_j}$. Here, $X_j^i$ denotes the input text, $Y_j^i$ represents the associated label for the $i$-th instance of the $j$-th task, and $M_j$ indicates the total number of samples in task $j$.

## 3.1 MODEL ARCHITECTURE

Our model architecture consists of the pre-trained language model $M_\theta$ as the shared backbone network, enhanced with an UnoLoRA$^\star$ adaptation module. This module is integrated into the self-attention and encoder-decoder attention sub-layers of both the encoder and decoder blocks in the transformer architecture, enabling task-specific adaptations without modifying the original pre-trained weights.

Given a pre-trained weight matrix $W \in \mathbb{R}^{d \times k}$, the UnoLoRA$^\star$-adapted weight matrix $W'$ is computed as:

$$W' = W + \alpha \cdot BA \tag{1}$$

where $B \in \mathbb{R}^{d \times r}$ and $A \in \mathbb{R}^{r \times k}$ are learnable low-rank matrices shared across all tasks, $r \ll \min(d, k)$ is the rank of the adaptation, with d being the input dimension and k being the output dimension, and $\alpha$ is a scaling factor (Figure 1).

Our PCA visualization comparing the distribution of LoRA matrices A and B in multi-task learning, as demonstrated in Figure 3, reveals that the $A$ matrix exhibits strong generalization capabilities across different tasks, while the $B$ matrix captures task-specific features. This finding motivated our design choice to multiply task-specific information with the $A$ matrix, leveraging its generalization power to better adapt to new tasks while maintaining task-specific knowledge.

To enable the model to distinguish between different tasks and learn task-specific adaptations within the shared LoRA space, we introduce a Shared Hypernetwork module. This module generates task-specific embeddings by combining task IDs, sample encodings, and position information:

$$e_t = H(t, s, p) \tag{2}$$

where $H$ is the Shared Hypernetwork, $t$ is the task ID, $s$ is the sample encoding, and $p$ is the position information. The output $e_t \in \mathbb{R}^{d_e}$ is a task-specific embedding, where $d_e$ is the dimensionality of the task embedding space (Figure 2). This unified embedding space is essential for enabling effective knowledge sharing across tasks while maintaining task-specific characteristics.

The inclusion of sample-level encodings is crucial as it allows the model to capture fine-grained, instance-specific features that may be relevant across multiple tasks. Layer-wise position embeddings provide important contextual information about how different layers in the transformer architecture process and transform the input, enabling more nuanced adaptations at different levels of abstraction. This multi-level representation ensures that the model can adapt its behavior based on both the specific requirements of each input sample and its position in the network hierarchy.

The Shared Hypernetwork consists of several components: A bottleneck network that processes the sample encodings:

$$b = B(s) \tag{3}$$

where $B$ is a multi-layer perceptron and $b \in \mathbb{R}^{d_b}$ is the bottleneck representation. This bottleneck architecture is crucial for distilling high-dimensional sample encodings into a compact, information-rich representation that captures essential features while reducing computational overhead.

Task and position embeddings:

$$e_{task} = E_{task}(t), \quad e_{pos} = E_{pos}(p) \tag{4}$$

where $E_{task}$ and $E_{pos}$ are embedding layers. These dedicated embedding spaces allow the model to learn distinct representations for task identity and structural position, ensuring that both task-specific requirements and architectural context are properly encoded.

A network that processes the concatenated position and task embeddings (Xiao et al., 2023) to differentiate between transformer blocks, and between the query and the value LoRA adapters:

$$e_t = C([e_{task}, e_{pos}, b]) \tag{5}$$

where $C$ is a multi-layer perceptron and $[\cdot, \cdot, \cdot]$ denotes concatenation. This fusion network is essential for learning complex interactions between task, position, and sample-specific information, creating a unified representation that captures all relevant aspects of the current adaptation context.

The task-specific embedding $e_t$ is then used to generate scaling factors that modulate the $A$ matrix in the UnoLoRA$^\star$ module:

$$s_t = S(e_t) \tag{6}$$

where $S : \mathbb{R}^{d_e} \to \mathbb{R}^r$ is a linear layer that projects the task embedding to the LoRA rank dimension. This projection layer plays a critical role in translating the rich task-specific information into appropriate scaling factors that can directly influence the LoRA adaptation process.

The task-specific scaling factors are applied to the LoRA adaptation:

$$W' = W + \alpha \cdot B(A \cdot \text{diag}(s_t)) \tag{7}$$

where $\text{diag}(s_t)$ creates a diagonal matrix from the scaling vector $s_t$. By applying the scaling to the $A$ matrix, we leverage its demonstrated generalization capabilities (Figure 3) while maintaining the task-specific adaptations learned by the $B$ matrix.

This approach provides a sophisticated mechanism for multi-task adaptation, combining the generalization power of the $A$ matrix with fine-grained task-specific information from the Shared Hypernetwork. The integration of sample-level encodings and position information enables the model to capture both instance-specific features and layer-wise contextual information, resulting in more effective and nuanced adaptations across different tasks.

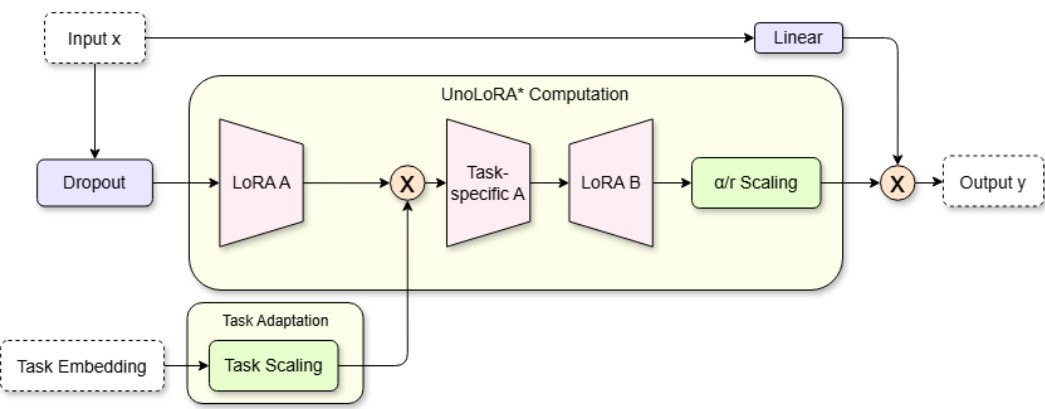

Figure 1: UnoLoRA$^\star$ Computation: Illustration of how UnoLoRA$^\star$ modifies the weight matrix $W$ using low-rank adaptation matrices $A$ and $B$. The scaling factor $\alpha$ and task-specific scaling vector $s_t$ allow for task-dependent adjustments.

## 3.2 TRAINING OBJECTIVE

We optimize the model parameters $\theta$, the shared LoRA matrices $B$ and $A$, the task embeddings $\{e_i\}$, and the hypernetwork using AdamW (Loshchilov & Hutter). The optimization process jointly learns the shared LoRA adaptation, task-specific scaling factors to maximize the overall performance across all tasks.

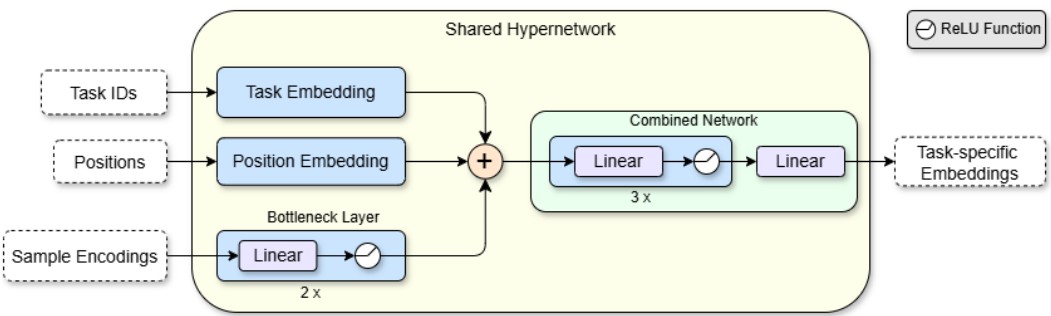

Figure 2: Shared Hypernetwork: The Shared Hypernetwork generates embeddings that encode task-specific information. These embeddings are used to adapt the LoRA weights dynamically, ensuring that task-specific nuances are captured effectively.

### 3.3 INTEGRATION WITH PRE-TRAINED MODELS

The integration of our UnoLoRA adapters with pre-trained language models is achieved through a wrapper architecture, which we call `EnhancedUnoloraWrapper`. This wrapper encapsulates a pre-trained T5 model and augments it with task-specific adaptations while preserving the original model's parameters. Given a pre-trained T5 model $M_\theta$, we replace specific linear layers in the self-attention and cross-attention modules with our UnoLoRA layers. The replacement occurs in both the encoder and decoder blocks:

$$M_\theta = \text{Replace}(M_\theta, \text{UnoLoRA}) \tag{8}$$

where Replace is a function that substitutes the query (Q) and value (V) projections in each attention layer with UnoLoRA modules. The UnoLoRA module extends the standard LoRA adaptation by incorporating task-specific scaling:

$$W't = W + \alpha \cdot (B \cdot \text{diag}(s_t))A \tag{9}$$

where $W't$ is the task-specific adapted weight matrix, $W$ is the original weight matrix, $B$ and $A$ are the LoRA matrices, $\alpha$ is the scaling factor, and $s_t$ is the task-specific scaling vector generated by the Shared Hypernetwork. The integration process involves the following steps:

1. Freezing the base model parameters:

$$\forall \theta \in M_\theta : \frac{\partial \mathcal{L}}{\partial \theta} = 0 \tag{10}$$

2. Replacing attention layers with UnoLoRA modules:

$$\text{AttnUnoLoRA} = \text{UnoLoRA}(\text{Attnoriginal}, r, \alpha, d_e) \tag{11}$$

where $r$ is the LoRA rank, $\alpha$ is the scaling factor, and $d_e$ is the task embedding dimension.

3. Initializing the Shared Hypernetwork:

$$H = \text{Shared Hypernetwork}(|\mathcal{T}|, d_h, d_e, d_s, d_b, L, p_{\max}) \tag{12}$$

where $|\mathcal{T}|$ is the number of tasks, $d_h$ is the hidden dimension, $d_e$ is the output dimension, $d_s$ is the sample encoding dimension, $d_b$ is the bottleneck dimension, $L$ is the number of layers, and $p_{\max}$ is the maximum position.

This integration approach allows for efficient task-specific adaptation of the pre-trained model while maintaining its original knowledge. The UnoLoRA modules and Shared Hypernetwork introduce a relatively small number of trainable parameters, enabling rapid adaptation to new tasks without the need for full model fine-tuning.

## 4 EXPERIMENTS

**Model and Implementation:**  We use T5-base (Raffel et al., 2020) as our backbone model across all experiments. For multi-task experiments, we implement UnoLoRA, a single custom version of LoRA (Hu et al., 2021) shared across all tasks. Furthermore, we implement UnoLoRA$^\star$ which is UnoLoRA with a shared hypernetwork to generate task specific embeddings to aid the LoRA learning process. For single-task experiments, we utilize the standard LoRA implementation from Hugging Face's Transformers (Wolf et al., 2020).

**Datasets and Evaluation:**  We evaluate our models on the GLUE (General Language Understanding Evaluation) benchmark (Wang, 2018), which includes tasks such as Natural Language Inference (MNLI), Sentiment Analysis (SST-2), Paraphrase Detection (MRPC, QQP), Textual Similarity (STS-B), Grammar Acceptance (CoLA), and Question Answering (QNLI, RTE), following the approach of Raffel et al. (2020). Since the original test sets are not publicly available, we adopt the data split strategy from Zhang et al. (2020). For smaller datasets (RTE, MRPC, STS-B, CoLA) with fewer than 10K samples, we split the original validation set equally into validation and test sets. For larger datasets, we create a validation set by reserving 1K samples from the training data and use the original validation set for testing.

**Baselines:**  We compare our method against several baselines, with careful consideration of how performance is measured and aggregated:

- **Single-Task Fine-Tuning:** Independent fine-tuning of T5-base for each task, updating all parameters.
- **Single-Task LoRA:** Independent LoRA adaptation for each task, resulting in separate task-specific adapters.The reported performance reflects the evaluation of a single task-specific adapter.
- **Multi-Task Fine-Tuning:** Simultaneous fine-tuning of T5-base on all tasks, updating all parameters.
- **HyperFormer++:** Implementation of the enhanced HyperFormer++ (Mahabadi et al., 2021) approach for multi-task learning with T5-base.

For single-task models, best checkpoint is taken for each task(one model per-task). For multi-task models, performance is measured using a single best checkpoint selected based on average validation performance across tasks.

**Experimental Details:**  All experiments use the GLUE benchmark's natural language understanding tasks. For multi-task training, we employ temperature-based sampling (T=10) to balance task representation. We train for 50 epochs on smaller datasets and 10 epochs on larger datasets during single-task LoRA fine-tuning. Following Raffel et al. (2020), we use a constant learning rate of $1e-4$ and train for $2^{18} = 262144$ steps, saving checkpoints every 29535 steps. Unlike Raffel et al. (2020), who report results using task-specific best checkpoints, we adopt a more realistic approach by selecting a single checkpoint based on the highest average validation performance across all tasks. This ensures fair comparison between single-task and multi-task approaches. Detailed hyperparameter settings are provided in Table 2 (See Appendix A.3). All experiments were conducted using NVIDIA A100 and H100 GPUs (40GB VRAM).

### 4.1 RESULTS ON THE GLUE BENCHMARK

We evaluate our proposed UnoLoRA method and its enhanced variant UnoLoRA$^\star$ against several baselines on the GLUE benchmark, with results presented in Table 1. Our analysis focuses on both performance and parameter efficiency across single-task and multi-task training paradigms. The experiments highlight the effectiveness of our approach in terms of parameter efficiency and its ability to leverage shared information across tasks through the shared LoRA adapter.

Table 1: Results on the GLUE benchmark. For MRPC and QQP, we report accuracy/F1. For STS-B, we report Pearson/Spearman correlation. For other tasks, we report the standard metric. Bold indicates best results in multi-task training. *Trained* is the per-task trainable parameters of the model. †: Results reported directly from Mahabadi et al. (2021).

| Model | #Params | | CoLA | SST-2 | MRPC | QQP | STS-B | MNLI | QNLI | RTE | Avg. |
|---|---|---|---|---|---|---|---|---|---|---|---|
| | Total | Trained | | | | | | | | | |
| *Single-Task Training* | | | | | | | | | | | |
| T5† (full fine-tuning) | 8.0× | 100% | 54.85 | 92.19 | **88.18 / 91.61** | **91.46 / 88.61** | 89.55 / 89.41 | **86.49** | 91.60 | 67.39 | **84.67** |
| LoRA | 1 + (8 × 0.004) | 0.4% | **56.10** | 93.81 | 84.31 / 84.31 | 90.44 / 89.84 | **90.19 / 89.79** | 86.29 | **93.56** | **69.78** | 84.40 |
| *Multi-Task Training* | | | | | | | | | | | |
| T5† (full fine-tuning) | 1.0× | 12.50% | 54.88 | 92.54 | **90.15 / 93.01** | **91.13 / 88.07** | 88.84 / 88.53 | 85.66 | 92.04 | 75.36 | 85.47 |
| HyperFormer++ | 1.02× | 0.290% | **63.73** | 94.03 | 89.66 / 92.63 | 90.28 / 87.20 | **90.00 / 89.66** | **85.74** | 93.02 | 75.36 | **86.48** |
| UnoLoRA *(Ours)* | 1.004× | **0.049%** | 50.79 | **94.61** | 85.78 / 85.78 | 89.99 / 89.32 | 88.63 / 88.44 | 84.71 | 92.68 | **77.70** | 84.40 |
| UnoLoRA⋆ *(Ours)* | 1.004× | 0.050% | 56.11 | 93.92 | 86.76 / 86.76 | 89.88 / 89.21 | 88.52 / 88.69 | 85.24 | **93.14** | 76.26 | 84.95 |

In the single-task setting, traditional LoRA achieves competitive performance (84.40% average) compared to full fine-tuning (84.67% average) while training only 0.4% of the parameters. This establishes a strong baseline for parameter-efficient fine-tuning approaches.

In the multi-task setting, our enhanced UnoLoRA⋆ achieves an average score of 84.95%, showing improvement over the base UnoLoRA's 84.40%. While HyperFormer++ achieves the highest average performance (86.48%), both our methods offer compelling parameter efficiency, using just a single shared adapter approach. UnoLoRA⋆ demonstrates particular strengths on several tasks, notably improving performance on CoLA (56.11% vs 50.79%), MRPC (86.76% vs 85.78%), and MNLI (85.24% vs 84.71%) compared to base UnoLoRA.

A key advantage of our approaches is their parameter efficiency. Both UnoLoRA and UnoLoRA⋆ require training only about 0.05% of the total parameters, significantly more efficient than full fine-tuning (12.50% in multi-task setting) and even HyperFormer++ (0.290%). This efficiency is particularly important in resource-constrained environments or when scaling to larger models.

Both variants show strong performance on classification tasks, with UnoLoRA excelling on SST-2 (94.61%) and UnoLoRA⋆ achieving strong results on QNLI (93.14%). Notably, UnoLoRA achieves the best RTE performance (77.70%) among all multi-task approaches, while UnoLoRA⋆ provides more consistent performance across the full task suite.

## 5 ANALYSIS

We present an analysis of UnoLoRA's performance and internal mechanisms, demonstrating its effectiveness in multi-task learning settings through three key findings: (1) superior parameter efficiency - as discussed earlier, (2) distinct functional specialization in the A and B matrices, and (3) faster convergence through our enhanced UnoLoRA⋆ variant.

### 5.1 EMPIRICAL ANALYSIS AND FINDINGS

To understand how UnoLoRA achieves efficient multi-task learning, we analyzed the properties of its LoRA matrices through multiple mathematical perspectives and discovered a clear functional specialization between components:

**Matrix Properties and Representations:** Our analysis reveals distinct characteristics between A and B matrices. The A matrices demonstrate higher singular values and eigenvalues (Figure 5c, 5d), indicating they capture a broader range of transformations in the parameter space. This is further supported by their scattered distribution in PCA visualization (Figure 3, left), suggesting diverse feature representations. In contrast, B matrices show more concentrated eigenvalue distributions and form dense clusters in PCA space (Figure 3, right), indicating more specialized transformations.

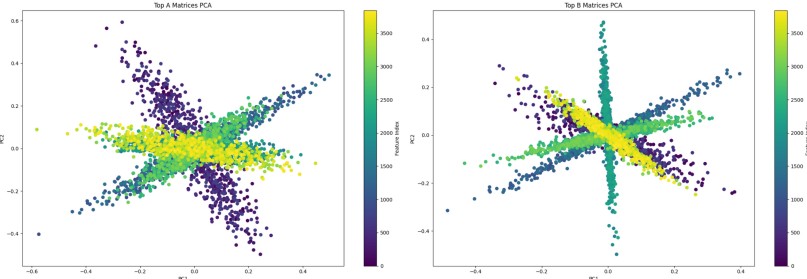

Figure 3: 2D PCA visualization comparing the distribution of LoRA matrices in multi-task learning. A matrices (left) exhibit a dispersed pattern with greater variance suggesting diverse, generalizable features. B matrices (right) show tighter clustering indicating task-specific feature specialization. Points represent individual matrix components projected onto the first two principal components.

**Layer-wise Behavior:**  Examining the cross-layer relationships, we observe that A matrices exhibit noticeable correlation across different layers (Figure 7, left), suggesting they learn consistent transformations throughout the network. This layer-wise generalization capability, combined with their diverse representational properties, makes them particularly suitable for multi-task learning. B matrices, conversely, show minimal cross-layer correlation (Figure 7, right), aligning with their role in task-specific adaptations.

This complementary behavior enables efficient multi-task learning through:

**Enhanced Representational Capacity:** Multi-task adaptations demonstrate consistently higher effective rank across all layers (Figure 5a), particularly pronounced in the encoder layers. As shown in Figure 4a, this enhanced representational capacity translates to superior performance scaling, where our multi-task approach (green dots) maintains efficiency across different parameter regimes compared to single-task models (red dots) and either matches or surpasses their performance.

**Optimal Parameter Updates:** The multi-task setting exhibits larger Frobenius norms (Figure 5b) and consistently higher singular values (Figure 5c) and eigenvalues (Figure 5d) compared to single-task counterparts. The larger Frobenius norms indicate stronger overall weight updates, suggesting the model makes more substantial adaptations to accommodate multiple tasks. The higher singular and eigenvalues reveal that these adaptations utilize a broader range of transformation directions in the parameter space, allowing the model to capture more complex patterns.

## 5.2 Convergence Analysis

Building on these insights, we developed UnoLoRA$^\star$, which enhances the base architecture's ability to learn task-specific features more efficiently. As demonstrated in Figure 4b, UnoLoRA$^\star$ achieves higher performance at earlier training stages compared to the original UnoLoRA. This pattern of faster convergence is consistently observed across multiple tasks in the GLUE benchmark, including MNLI, STS-B, QQP, and SST-2 (see Appendix A.1 for detailed per-task convergence plots). This faster convergence is particularly valuable in resource-constrained scenarios and rapid deployment settings.

The improved early-stage performance can be attributed to the enhanced architecture's ability to better leverage the functional specialization we observed between A and B matrices, allowing for more efficient learning of both shared and task-specific features.

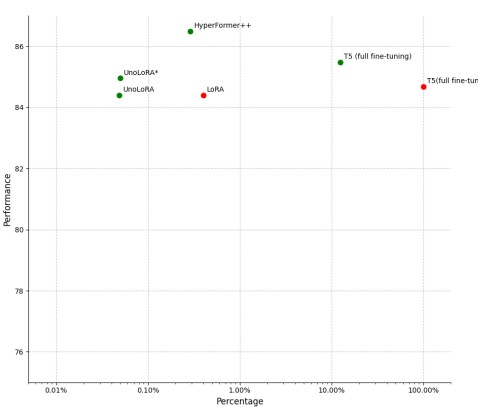

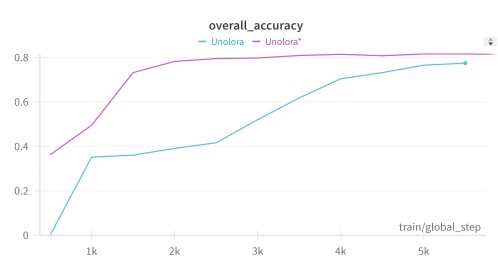

(a) Comparison of accuracy versus percentage (log scale) of trained parameters across different models in a multi-task learning setting. The green dots represent multi-task fine-tuned models, and the red dots represent single-task fine-tuned models.

(b) Comparison of the overall accuracy of UnoLoRA and UnoLoRA⋆ over the first 5000 training steps on a subset of the validation dataset. We can see that UnoLoRA⋆ is able to achieve a higher performance at an earlier stage than UnoLoRA.

Figure 4: Performance analysis of multi-task learning models. (a) Illustrates the trade-off between model accuracy and the percentage of trained parameters across various models. (b) Shows the per-step convergence rates of UnoLoRA and UnoLoRA⋆, highlighting the improved early-stage performance of UnoLoRA⋆.

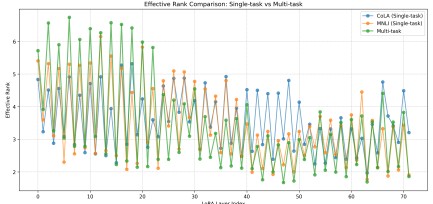

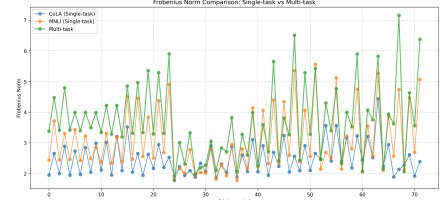

(a) Effective rank comparison across model layers, showing consistently higher values for multi-task training.

(b) Layer-wise Frobenius norm distribution, indicating magnitude of weight adjustments in adaptation matrices.

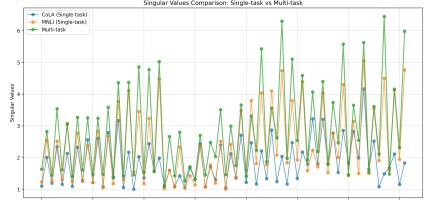

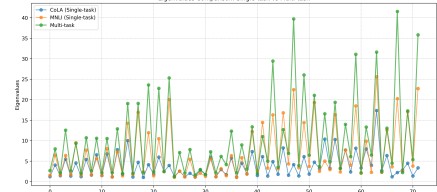

(c) Distribution of singular values across layers, reflecting the complexity of learned transformations.

(d) Eigenvalue distributions showing the dimensionality of learned feature spaces.

Figure 5: Comparative analysis of LoRA adaptation matrices between single-task (CoLA, MNLI) and multi-task models across different metrics. Layer indices (x-axis) correspond to the model architecture progression from encoder (lower indices) to decoder (higher indices). The multi-task model consistently demonstrates higher effective rank and more distributed eigenvalue patterns, suggesting more complex and comprehensive feature representations compared to single-task variants. This analysis spans multiple mathematical perspectives: effective rank (measuring dimension utilization), Frobenius norm (capturing overall adaptation magnitude), and spectral properties (singular and eigenvalues) revealing the internal structure of learned transformations.

## 6 LIMITATIONS AND FUTURE WORK

While UnoLoRA demonstrates promising results in multi-task learning with parameter-efficient fine-tuning, several limitations and opportunities for future research remain. Firstly, our evaluation mainly focuses on the GLUE (Wang, 2018) benchmark. While the dataset is comprehensive, evaluating UnoLoRA on additional datasets would further reinforce the results obtained.

Our experiments have been conducted exclusively with the T5-base model (Raffel et al., 2020), which uses an encoder-decoder architecture. Future work could investigate UnoLoRA's effectiveness with other architectural paradigms, such as encoder-only (e.g., BERT) and decoder-only (e.g., GPT) models. Additionally, testing UnoLoRA across different model scales, both smaller and larger, would provide valuable insights into its scalability and efficiency characteristics.

Several promising directions emerge for future research. First, investigating UnoLoRA's performance in few-shot learning scenarios would help understand its effectiveness with limited training data. Second, exploring task transfer capabilities, particularly between unrelated domains, would provide insights into cross-domain generalization. Third, extending UnoLoRA beyond natural language processing to other modalities such as computer vision and audio processing would evaluate its broader applicability. Furthermore, the shared nature of our single LoRA module presents unique opportunities for interpretability research. Unlike models with separate task-specific modules, our approach could enable better analysis of how different tasks influence the learned weights, potentially providing insights into task relationships and knowledge transfer mechanisms. These extensions would help establish the boundaries of UnoLoRA's capabilities and potentially reveal new applications for parameter-efficient multi-task learning.

## 7 CONCLUSION

This paper introduces UnoLoRA, demonstrating that a single shared LoRA module can effectively handle multi-task learning while requiring only 0.05% trainable parameters per task. Unlike traditional approaches that require separate LoRA modules for each task, our approach only requires one LoRA module for all tasks, and achieves competitive performance on the GLUE benchmark.

We further enhance this architecture with UnoLoRA$^\star$, which employs a shared hypernetwork to generate task-specific embeddings. This enhancement leads to significantly faster convergence across multiple GLUE tasks, making it particularly valuable for resource-constrained scenarios and rapid deployment settings. Our empirical analysis reveals how UnoLoRA achieves efficient multi-task learning through complementary roles of its components: A matrices capture generalizable features with consistent cross-layer transformations, while B matrices handle task-specific adaptations.

These findings establish UnoLoRA as a promising PEFT method for multi-task learning. The success of our approach in maintaining performance while drastically reducing parameters opens new possibilities for efficient model adaptation and has the potential to inspire further research in PEFT methods, particularly in scenarios where computational resources are limited.

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

## A APPENDIX

### SUPPLEMENTARY MATERIAL

Please find the link attached to replicate our experiments: Google Drive Link

## A.1 TASK CONVERGENCE PLOTS

To investigate how UnoLoRA*'s hypernetwork affects training dynamics, we compare learning curves against the base UnoLoRA model across four representative tasks from the GLUE benchmark. Figure 6 presents the convergence plots for STS-B, SST-2, MNLI, and QQP, showcasing tasks with varying complexity and objectives.

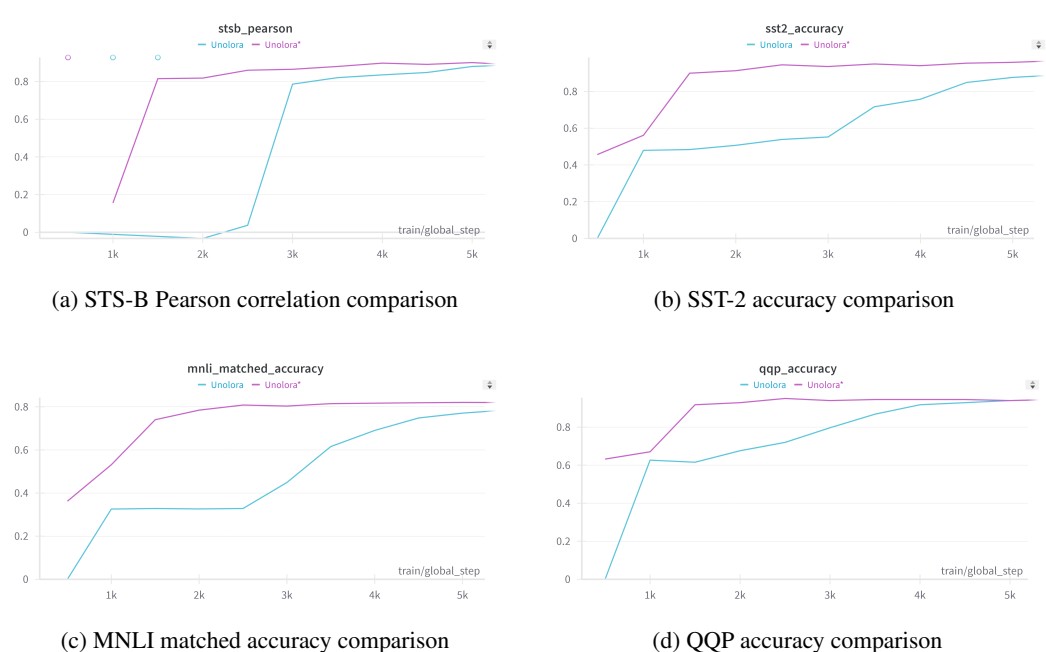

(a) STS-B Pearson correlation comparison

(b) SST-2 accuracy comparison

(c) MNLI matched accuracy comparison

(d) QQP accuracy comparison

Figure 6: Convergence plots for (a) STS-B Pearson correlation, (b) SST-2 accuracy, (c) MNLI matched accuracy, and (d) QQP accuracy. Each plot compares the performance of UnoLoRA and UnoLoRA* over training steps. UnoLoRA* consistently demonstrates faster convergence and better early-stage performance across all tasks, regardless of the task complexity or evaluation metric.

The plots demonstrate that UnoLoRA*'s improved convergence is not task-specific but rather a general characteristic of the enhanced architecture. This is particularly evident in more complex tasks like MNLI (natural language inference) and QQP (semantic similarity), where the performance gap between UnoLoRA* and the base UnoLoRA is more pronounced in the early stages of training. Even for simpler tasks like SST-2 (sentiment analysis), UnoLoRA* maintains its advantage in convergence speed while achieving comparable final performance.

## A.2 LoRA Matrices Correlation Across Layers

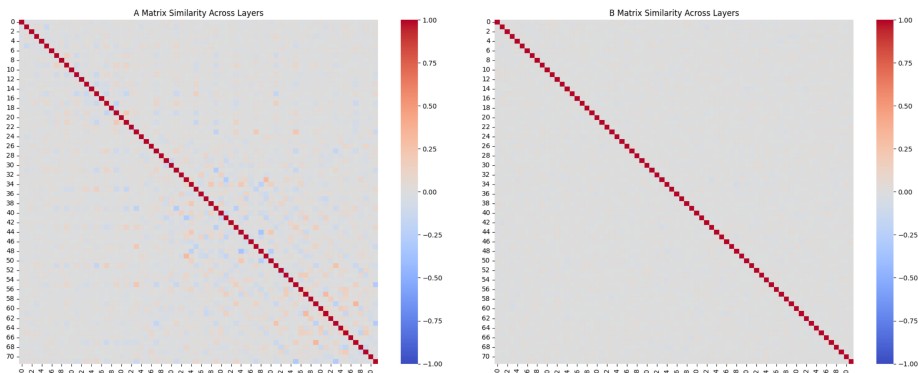

Figure 7: Pearson correlation analysis of LoRA matrices across network layers in multi-task learning. Left: A matrices show noticeable correlation between different layers suggesting these matrices learn similar transformations across layers. This consistency across layers indicates the learning of general features that are reused throughout the network, supporting their role in capturing transferable knowledge. Right: B matrices show minimal correlations between layers implying each layer learns distinct transformations, consistent with their role in capturing task-specific adaptations.

## A.3 Sampling Strategies for Multi-Task Learning

In multi-task learning, the sampling strategy plays a crucial role in determining the proportion of data from each task that the model is trained on. The goal is to strike a balance between providing enough data for the model to learn each task effectively while avoiding over-training on any particular task. Several sampling strategies have been proposed to address this challenge:

**Examples-Proportional Mixing:** This strategy samples examples from each task in proportion to the size of its dataset. It is equivalent to concatenating all the datasets and randomly sampling examples from the combined dataset. However, when there is a significant disparity in dataset sizes, such as the inclusion of a large unsupervised task, this approach can lead to under-training on the supervised tasks. To mitigate this issue, an artificial "limit" can be set on the dataset sizes before computing the proportions.

**Temperature-Scaled Mixing:** Temperature scaling is another way to address the imbalance in dataset sizes. In this approach, the mixing rates of each task are raised to the power of the reciprocal of a temperature parameter $T$ and then renormalized. When $T = 1$, it is equivalent to examples-proportional mixing. As $T$ increases, the mixing proportions become closer to equal mixing. This allows for adjusting the influence of larger datasets while still considering their relative sizes. The `MultiTaskBatchSampler` used in Mahabadi et al. (2021) falls under this category of temperature-scaled mixing. It aims to balance the proportions of tasks in each batch by sampling tasks according to their dataset sizes. However, this approach can still lead to oversampling of smaller datasets like RTE, as the proportions are solely based on the dataset sizes without considering other factors such as task difficulty or model performance.

During multi-task training, we sample tasks with conventional temperature-based sampling, using a temperature of $T = 10$, following . Tasks are sampled proportionally to $p_\tau^{1/T}$, where $p_\tau = \frac{N_\tau}{\sum_{i=1}^{T} N_\tau}$ and $N_\tau$ is the number of training samples for the $\tau$-th task.

**Equal Mixing:** In this strategy, examples are sampled from each task with equal probability, regardless of the dataset sizes. While this ensures equal representation of all tasks, it may lead to overfitting on low-resource tasks and underfitting on high-resource tasks.

## A.4 HYPERPARAMETERS

Table 2: Hyperparameter settings of T5-base models on GLUE for UnoLoRA and UnoLoRA$^{\star}$

| Hyperparameters | MNLI | SST-2 | MRPC | CoLA | QNLI | QQP | RTE | STS-B |
|---|---|---|---|---|---|---|---|---|
| Rank $r$ | | | | 8 | | | | |
| Alpha | | | | 16 | | | | |
| Layer $L$ | | | All Q,V Self-Attention | | | | | |
| Bottleneck dim $L$ | | | | 8 | | | | |
| Sample encoding dim $L$ | | | | 512 | | | | |
| Dropout | | | | 0.1 | | | | |
| Optimizer | | | | AdamW | | | | |
| Learning Rate | | | | 1e-4 | | | | |
| Weight decay | | | | 0.01 | | | | |
| Warmup steps | | | | 1000 | | | | |
| Max steps | | | | 262144 | | | | |
| LR scheduler | | | Cosine Annealing | | | | | |
| Batch size | | | | 32 | | | | |
| Epochs | 10 | 10 | 50 | 50 | 10 | 10 | 50 | 50 |

