# OpenReview forum: "UnoLoRA: Single Low-Rank Adaptation for Efficient Multitask Fine-tuning"
_ICLR.cc/2025/Conference — ICLR 2025 Conference Withdrawn Submission_

### Official Review · Reviewer_LP9b · 2024-10-31

**Soundness:** 3
**Presentation:** 2
**Contribution:** 2
**Rating:** 3
**Confidence:** 3

**Summary:**

This article proposes a new method called UNOLORA, which utilizes shared low-rank adaptation (LoRA) modules to achieve efficient multi-task learning for large language models, and has achieved outstanding performance on the GLUE benchmark.

**Strengths:**

- The method proposed by the authors is simple but effective.

**Weaknesses:**

- The writing and presentation is not good, for example, the caption and figure of Figure 1 seems confusing. Also the font size in the figure is too small to understand.

- The training of Shared Hypernetwork will introduce additional training cost.

- The method is only evaluated on one model, without scaling up the model size/architecture.

**Questions:**

- What is the difference between the UNOLora* and UNOLoRA? I haven't found the method difference in your paper?

- It required a comparation to use LoRA to multi task training.

- It is not clear why cross task relation is related to the capability of using LoRA to do multi-task learning.

---

> ### Author Response · Authors · 2024-12-04
>
> Dear Reviewer LP9b,
>
> We thank you for your detailed feedback and thoughtful comments on our work. We first answer the questions you had asked:
>
> — What is the difference between UnoLoRA and UnoLoRA*?
>
> For UnoLoRA, we developed a custom implementation that extends a single base LoRA for multi-task scenarios. UnoLoRA* is an enhanced version of UnoLoRA that incorporates a shared hypernetwork to generate task-specific embeddings. These embeddings dynamically scale the LoRA matrices, enabling fine-grained task-specific adaptations. This addition improves convergence and ensures consistent performance across tasks, as highlighted in our analysis.
>
> — It is not clear why cross-task relation is related to the capability of using LoRA to do multi-task learning:
>
> The fundamental evidence for cross-task learning comes from the parameter efficiency itself: maintaining performance while drastically reducing per-task parameters is possible because the model is effectively sharing knowledge across tasks. This is also supported by our analysis of different indicators(Fig. (3) and Fig. (5)) which suggests how the LoRA matrices may function: A captures general features, while B handles task-specific adaptations. This sharing mechanism inherently leverages cross-task relationships, enabling the model to generalize effectively across tasks while minimizing per-task parameters.
>
>
> We also appreciate your constructive feedback and address the specific weaknesses called out:
>
> - We acknowledge that Figure 1 could have been clearer. In the revised manuscript, we will be improving the figure’s readability by increasing the font size, ensuring all elements are clearly labeled, and refining the caption to align with the textual description.
>
> - While the Shared Hypernetwork introduces slight additional costs, these are minimal. UnoLoRA requires 0.049% per-task parameters, compared to 0.050% for UnoLoRA*. Similarly, UnoLoRA* takes only slightly longer to train (24.8 hours for UnoLoRA vs. 25.8 hours for UnoLoRA*) but achieves near-peak performance in significantly fewer steps (Fig. 4(b)). This makes UnoLoRA* highly efficient given its improved convergence and performance.
>
> - We acknowledge the limitation of evaluating UnoLoRA only on the T5-base model. To address this, we are actively extending UnoLoRA to other architectures, such as Llama-3, to validate its scalability and performance across diverse model configurations.
>
>
> We thank you again for your valuable feedback, which will help us significantly improve the clarity of our work. We are happy to provide any additional details.

---

### Official Review · Reviewer_7i5S · 2024-10-31

**Soundness:** 1
**Presentation:** 1
**Contribution:** 2
**Rating:** 3
**Confidence:** 4

**Summary:**

This paper presents a method called UnoLoRA, a procedure for constructing
low-rank Transformer adapters in a multi-task setting by training a network to
apply task-specific transformations to a shared adapter. In particular, while
standard LoRA parameterizes weight matrices as $W + AB^\top$ for low-rank
$A$ and $B$, UnoLoRA parameterizes them as $W + A ~\mathrm{diag}(H(t)) ~ B^\top$,
where $t$ is a task representation that includes both a discrete identifier
and example data and positional embeddings, and $H$ is a hypernetwork. A similar recipe was previously
explored by Karimi Mahabadi et al. (2021) under the name of "HyperFormers"; as
far as I can tell, the main differences are that:

- HyperFormers condition only on task IDs, while UnoLoRA conditions on example
  input data

- HyperFormers also modulate LayerNorm parameters, and not just adapters

- HyperFormers use a slightly different adapter parameterization from the modern LoRA recipe, with
  a nonlinearity in the middle

**Strengths:**

- Simple and seemingly effective way of parameterizing low-rank adapters in the multitask setting. The idea is timely---there have been a lot of improvements in LoRA and related schemes in the last couple of years, and revisiting conditional computation + adapter combinations seems like a promising direction.

**Weaknesses:**

- Comparatively minor tweak of an existing idea. This wouldn't be an issue on
  its own, except for the fact that the various changes are not evaluated in
  a way that enables direct comparison to HyperFormers, as described below.

- Inconsistencies and missing details in the description of the method. Fig 1
  makes reference to a "Task-specific A" parameter that is not mentioned
  anywhere in the formal description of the method---is it used, and if so,
  where? Additionally, the experiments make reference to a method called
  UnoLoRA$^*$, which achieves slightly better performance than the base method
  but does not appear to be described anywhere.

- Major issues in evaluation. The paper's main results are summarized in Fig
  6(a), which show that UnoLoRA and HyperFormers both pareto-dominate training
  separate adapters for each task---UnoLoRA involves fewer parameters at the
  same level of performance, while HyperFormers give increased accuracy but are
  slightly less parameter-efficient than UnoLoRA. I have two concerns here.

    - First, the individual differences between UnoLoRA and HyperFormers are
      never individually evaluated, making it impossible to figure which (if any)
      are responsible for the performance differences.

    - Second, and more fundamentally---the whole point of adapter-based methods
      is that they provide a tunable parameter (the adapter rank) that trades
      off between accuracy and parameter count. So what we really need to see
      is the entire accuracy / efficiency curve for both model classes, rather
      than an arbitrary point on each. In fact, if I understand correctly,
      even the size of the adapter is totally incomparable between the two
      models being compared: this paper trains UnoLoRA with a rank of 8, while
      the results copied from the HyperFormers paper appear to use a rank of 24.

  Without a minimal comparison (or a complete frontier from each model), it is
  possible that all observed differences between methods result from
  incomparable hyperparameter choices.

- Major formatting issues: nearly every citation in the paper is incorrectly formatted (using \citet instead of \citep). It seems likely that this paper didn't receive even a single round of proofreading, and should not have been submitted to ICLR in its current form.

**Questions:**

- How does performance change as rank is varied?
- How do individual components of the method affect performance?
- What is UnoLoRA$^*$?
- Is there a task-specific $A$ matrix or not?

---

> ### Author Response · Authors · 2024-12-04
>
> Dear Reviewer 7i5S,
>
> We thank you for your detailed feedback and comments on our work. We first answer the questions you had asked:
>
>
> — How does performance change as rank is varied?
>
> We are in the process of comparing the performance with different ranks.
>
> — How do individual components of the method affect performance?
>
> 1. Shared LoRA Adapters: This is the main cause of parameter efficiency by enabling multi-task learning with a single adapter across tasks.
>
> 2. Task-Specific Scaling (UnoLoRA*): Allows the shared adapter to dynamically adjust for task-specific nuances, improving task discrimination and leads to faster convergence to peak performance.
>
> While the graphs in the Analysis section of the paper provide some good empirical insights into this, we are performing a thorough ablation study on our architecture to prove the contribution of the individual components quantitatively.
>
> — What is UnoLoRA*?
>
> For UnoLoRA, we developed a custom implementation that extends a single base LoRA for multi-task scenarios. UnoLoRA* is an enhanced version of UnoLoRA that incorporates a shared hypernetwork to generate task-specific embeddings. These embeddings dynamically scale the LoRA matrices, enabling fine-grained task-specific adaptations. This addition improves convergence and ensures consistent performance across tasks, as highlighted in our analysis.
>
> — Is there a task-specific A matrix or not?
>
> No, there is no dedicated task-specific A matrix. Instead, the A matrix is shared across tasks, and task-specific A matrices are dynamically generated by scaling this shared A matrix with task-specific scaling factors produced by the shared hypernetwork.
>
> We also appreciate your constructive feedback, and address the specific weaknesses called out:
>
>
> - The description of UnoLoRA*, which integrates a shared hypernetwork for task-specific embeddings, has been clarified with more details in our edit.
>
> - We do not understand the point on comparison with the results from the HyperFormer paper (especially the rank being 24 - where do we find this?)
>
> - Lastly, we regret the citation formatting issues (\citet vs. \citep) in the initial submission; these have been thoroughly corrected in the revised manuscript.
>
>
> We thank you again for your valuable feedback, which will help us significantly improve the clarity of our work and directions for future experiments. We are happy to provide any additional details.

---

### Official Review · Reviewer_JqKR · 2024-11-03

**Soundness:** 2
**Presentation:** 1
**Contribution:** 2
**Rating:** 3
**Confidence:** 3

**Summary:**

The paper introduces UnoLoRA, a method for parameter-efficient multitask fine-tuning of large language models (LLMs) through a shared Low-Rank Adaptation (LoRA) module. UnoLoRA leverages LoRA's implicit regularization properties to facilitate multitask learning by using a single adapter shared across all tasks, instead of separate adapters for each task. This approach drastically reduces trainable parameters to 0.05% per task while maintaining competitive performance with existing multitask methods. The model is evaluated on the GLUE benchmark and demonstrates parameter efficiency and improved generalization by capturing both shared and task-specific information. The authors further refine their method with UnoLoRA⋆, which converges faster and performs better in early training stages compared to the initial UnoLoRA.

**Strengths:**

- The authors conduct in-depth analyses of LoRA matrices in both single-task and multitask settings, highlighting distinctions in their properties (like effective rank and Frobenius norm) and the roles of A and B matrices. Visualizations like PCA further illustrate how UnoLoRA efficiently manages task-shared and task-specific information.
- The study’s experiments on the GLUE benchmark provide extensive evidence of UnoLoRA's effectiveness and competitive performance.

**Weaknesses:**

- For the experiments on the GLUE benchmark, no repeated experiments with different random seeds were performed, and the experimental results are not completely convincing due to the randomness.
- Only the T5-base model was used for the experiment. The effectiveness of the method was not verified on larger or smaller models, nor on decoder-only models.

**Questions:**

- What is the relationship between Figure 2 and Figure 1? Which part of Figure 1 is the Shared Hypernetwork shown in Figure 2?
- For different tasks, does UnoLoRA only change the task embedding and keep the other parts shared between different tasks?

---

> ### Author Response · Authors · 2024-12-04
>
> Dear Reviewer JqKR,
>
> We thank you for your detailed feedback and thoughtful comments on our work. We first answer the questions you had asked:
>
> — What is the relationship between Figure 2 and Figure 1? Which part of Figure 1 is the Shared Hypernetwork shown in Figure 2?
>
> Figure 2 illustrates the Shared Hypernetwork architecture, which generates task-specific embeddings by processing task IDs, layer-wise intermediary position embeddings, and the encoded representation of the input sample. The relationship between Figures 1 and 2 lies in the role of the task embedding: the output of the Shared Hypernetwork in Figure 2 is provided as input to the UnoLoRA* computation shown in Figure 1. In this setup, the hypernetwork produces the task embedding dynamically, which is then used to scale the shared LoRA matrices A in Figure 1 for task-specific adaptations.
>
> — For different tasks, does UnoLoRA* only change the task embedding and keep the other parts shared between different tasks?
>
> Yes, UnoLoRA* maintains shared parameters across tasks while only varying task-specific embeddings. Here’s how this works:
>
> 1. Shared Components: The main LoRA matrices (lora_A and lora_B) are shared across all tasks.
>
> 2. Task-Specific Adaptation: Task-specific embeddings and their associated scaling factors vary between tasks, allowing task-specific behavior.
>
> 3. Dynamic Task Adaptation: The task-specific embeddings generated by the Shared Hypernetwork dynamically scale the shared LoRA A matrices, ensuring parameter efficiency while maintaining task-specific adaptations.
>
> This design ensures that the majority of parameters are shared, with only lightweight task embedding components varying across tasks, enabling UnoLoRA* to achieve both efficiency and flexibility.
>
>
> We also appreciate your constructive feedback and address the specific weaknesses called out:
>
> - We acknowledge that the experimental results were not repeated with multiple random seeds, which could have added robustness to our findings. In future revisions, we will perform additional runs with varying random seeds to quantify the variability and report averaged results, ensuring stronger statistical reliability.
>
> - We agree that evaluating UnoLoRA solely on T5-base limits its generalizability. We are actively extending our method to other architectures, including larger and smaller models, as well as decoder-only models like Llama-3, to validate its scalability and broader applicability. These experiments are ongoing, and we will include results in subsequent updates.
>
>
> We thank you again for your insightful feedback, which has been invaluable in improving the clarity of our work. We are happy to provide any additional details.

---

### Official Review · Reviewer_zcEE · 2024-11-04

**Soundness:** 2
**Presentation:** 2
**Contribution:** 2
**Rating:** 3
**Confidence:** 4

**Summary:**

The paper presents UnoLoRA, an approach for parameter-efficient multitask learning in large language models (LLMs) using a single Low-Rank Adaptation (LoRA) module shared across multiple tasks. Building upon LoRA as an implicit regularizer, the authors explore its application in a multitasking context, aiming to reduce the number of trainable parameters while maintaining competitive performance. The paper introduces an architecture, UnoLoRA, which integrates a shared hypernetwork that generates task-specific scaling factors.

**Strengths:**

- The paper conducts comprehensive experiments and analysis to verify the proposed method.
- The paper is well structured, proposing an architecture, UnoLoRA, which integrates a shared hypernetwork that generates task-specific scaling factors.

**Weaknesses:**

- The experiments are conducted on T5-series models, which are from 4 years ago. Using a more recent model doesn't necessarily mean aiming for the current SOTA (state-of-the-art), but rather that the behaviors of stronger models might differ, making experiments on T5 impractical. For instance, current models, after instruction tuning, demonstrate strong zero-shot generalization across tasks, making multi-task learning less important.
- In the first table, the method proposed in this paper does not outperform HyperFormer++, even though they have different amounts of training parameters, the average effectiveness is also quite lacking. Therefore, the experimental results of this paper are not very convincing.

**Questions:**

- Why not use a self-implemented LoRA in both multi-task and single-task scenarios, since LoRA is relatively simple to implement?
- Is there a detailed efficiency analysis available?
-  How to acquire the task embeddings in the paper?

---

> ### Author Response · Authors · 2024-12-04
>
> Dear Reviewer zcEE
>
> We thank you for your detailed feedback and thoughtful comments on our work. We will first take up the questions you had asked:
>
>
> — Why not use a self-implemented LoRA in both multi-task and single-task scenarios, since LoRA is relatively simple to implement?
>
> Our work implements both a standard LoRA baseline and our proposed UnoLoRA method. This dual implementation allows for direct comparisons under identical training conditions. While LoRA is relatively simple to implement, we utilized the PEFT library's implementation for the baseline to ensure comparisons against a well-tested, community-standard version. For UnoLoRA, we developed a custom implementation that extends a single base LoRA for multi-task scenarios. UnoLoRA* is an extension of UnoLoRA that uses shared hypernetworks to generate task-specific embeddings to scale the A-matrix in LoRA
>
>
> — Is there a detailed efficiency analysis available?
>
> So far, we have conducted an efficiency analysis focusing on parameter usage and computational cost. UnoLoRA requires only 0.049% trainable parameters per task, while UnoLoRA⋆ requires 0.050%. Despite the slight increase, UnoLoRA⋆ achieves faster convergence and improved performance in the early stages of training. In terms of runtime, UnoLoRA⋆ takes only marginally longer (24.8 hours for UnoLoRA vs. 25.8 hours for UnoLoRA⋆ on the same setup). This makes UnoLoRA* very effective given that it reaches near-peak performance in much fewer steps(Fig. 4(b)).
>
> — How are task embeddings acquired in the paper?
>
> In our approach, task embeddings are generated through a principled three-component mechanism:
>
> 1. Task Identification: Unique task IDs provide high-level task-specific context.
>
> 2. Contextual Information: Encoded representations of input tokens capture instance-specific features.
>
> 3. Positional Information: Layer-wise intermediary embeddings capture structural information for task-specific modulation.
>
> These components are processed through a shared hypernetwork, which generates task-specific scaling factors. This dynamic adaptation mechanism maintains parameter efficiency while enabling effective task-specific behavior.
>
>
> We also appreciate your constructive feedback and address the specific weaknesses called out:
>
> - We acknowledge the limitation of using T5-series models, which are not the most recent architecture. However, T5 was chosen to maintain consistency with prior work, such as HyperFormer, enabling direct and fair comparisons. This ensures that improvements stem from our methodology rather than discrepancies in model architectures. That said, we are actively extending UnoLoRA to newer architectures, including decoder-only models like LLaMA-3, to evaluate its generalizability and relevance in contemporary contexts.
>
> - The reviewer notes that our method does not outperform HyperFormer++ in some metrics. While HyperFormer++ performs slightly better in many tasks, UnoLoRA* provides superior parameter efficiency and faster convergence. Additionally, our results demonstrate UnoLoRA*’s competitive performance at significantly lower parameter costs(1/6th of HyperFormer++). We are conducting further experiments, including ablation studies and rank-matched comparisons, to provide a more comprehensive evaluation.
>
>
> We thank you again for your insightful feedback, which has been invaluable in improving the clarity of our work. We are happy to provide any additional details.

---

### Note · Authors · 2024-12-04

I have read and agree with the venue's withdrawal policy on behalf of myself and my co-authors.